# NanoSafe III: A User Friendly Safety Management System for Nanomaterials in Laboratories and Small Facilities

**DOI:** 10.3390/nano11102768

**Published:** 2021-10-19

**Authors:** Elina Buitrago, Anna Maria Novello, Alke Fink, Michael Riediker, Barbara Rothen-Rutishauser, Thierry Meyer

**Affiliations:** 1Ecole Polytechnique Fédérale de Lausanne (EPFL), Occupational Health and Safety (OHS), Station 6, CH-1015 Lausanne, Switzerland; elina.buitrago@epfl.ch (E.B.); anna.novello@epfl.ch (A.M.N.); 2BioNanomaterials, Adolphe Merkle Institute, University of Fribourg, Ch. des Verdiers 4, CH-1700 Fribourg, Switzerland; alke.fink@unifr.ch (A.F.); barbara.rothen@unifr.ch (B.R.-R.); 3SCOEH: Swiss Centre for Occupational and Environmental Health, Binzhofstrasse 87, CH-8404 Winterthur, Switzerland; michael.riediker@alumni.ethz.ch; 4Ecole Polytechnique Fédérale de Lausanne (EPFL), Group of Chemical and Physical Safety (ISIC-GSCP), Station 6, CH-1015 Lausanne, Switzerland

**Keywords:** engineered nanomaterials, risk management, nanosafety, occupational exposure, control banding, safe handling practices

## Abstract

Research in nanoscience continues to bring forward a steady stream of new nanomaterials and processes that are being developed and marketed. While scientific committees and expert groups deal with the harmonization of terminology and legal challenges, risk assessors in research labs continue to have to deal with the gap between regulations and rapidly developing information. The risk assessment of nanomaterial processes is currently slow and tedious because it is performed on a material-by-material basis. Safety data sheets are rarely available for (new) nanomaterials, and even when they are, they often lack nano-specific information. Exposure estimations or measurements are difficult to perform and require sophisticated and expensive equipment and personal expertise. The use of banding-based risk assessment tools for laboratory environments is an efficient way to evaluate the occupational risks associated with nanomaterials. Herein, we present an updated version of our risk assessment tool for working with nanomaterials based on a three-step control banding approach and the precautionary principle. The first step is to determine the hazard band of the nanomaterial. A decision tree allows the assignment of the material to one of three bands based on known or expected effects on human health. In the second step, the work exposure is evaluated and the processes are classified into three “nano” levels for each specific hazard band. The work exposure is estimated using a laboratory exposure model. The result of this calculation in combination with recommended occupational exposure limits (rOEL) for nanomaterials and an additional safety factor gives the final “nano” level. Finally, we update the technical, organizational, and personal protective measures to allow nanomaterial processes to be established in research environments.

## 1. Introduction

Engineered nanomaterials (ENM) differ from their corresponding bulk materials in size and surface area, which potentially impacts their chemical, physical, and biological properties. Nanotechnology is a fast-growing research field that offers solutions for many problems. For example, nanomaterials are already used in food technology, medicine and medical devices, composite materials, and textiles, and the list is getting longer every year. Because of the technological advancements nanotechnology has brought about, exposure to ENM is predicted to increase in the future. The European commission estimated the nanomaterial sector to have produced 11 million tons of products at a market value of €20 billion in 2016, with between 300,000 and 400,000 people employed in the sector [1]. An updated report published in 2017 predicted a market growth rate of 18.2% from 2016 to 2021 [2]. With the increases in research and the production rate, more workers will be exposed to ENM at their workplace, especially during research, production, and maintenance. Occupational safety specialists are, therefore, faced with the challenging task of protecting workers from possible risks involving ENM with unknown hazards. Regulatory bodies have responded to the needs of industry, with guidelines and regulations being steadily updated to include specific guidelines for ENM.

In 2020, REACH extended its regulations to include explicit requirements for companies that manufacture nanomaterials that include specific information requirements [3]. 

The OECD test guideline programme and programme on manufactured nanomaterials have collaborated to develop standardized methods to generate reliable data [4]. These guidelines are meant to assist in the derivation of necessary information for the safety assessment of ENM. 

The WHO has published guidelines for protecting workers from potential risks of ENM, which cover hazard assessment, control measures for exposure from inhaled ENM, as well as recommendations for training of workers [5]. The guidelines recommend grouping of similar materials across hazard categories. The working group also concludes that more research is needed to identify biomarkers to measure exposure and long-term adverse health effects. 

Even with these guidelines and standards for the safety assessment of ENM, the research and development phase lacks relevant safety data. 

Hazard identification for ENM is a long and tedious process, which requires the generation of large amounts of data that are often not available in the initial stage of development, and even if they are, require substantial literature searching. This issue requires proper Occupational Exposure Limits (OEL) to be established. According to the ECHA, OEL “are regulatory values that indicate levels of exposure that are considered to be safe (health-based) for a chemical substance in the air of a workplace” [6]. They are used as reference values for assessing and controlling workplace exposure, determining the need for personal protective equipment (PPE), and implementing medical surveillance. Ideally, to determine proper OEL, it would be necessary to have a full set of dose–response data from animals and human studies, estimates of adverse health risks in workers based on quantitative risk assessments, together with workplace exposure and control data. Due to the great variety of ENM, this is not possible in practice [7,8]. To date, no OEL has been established for ENM; however, many regulatory agencies are starting to make lists of recommended exposure limits for the most used ENM, for which a good selection of hazard and exposure data already exist (see Appendix A). These values can then be used as references for other ENM with similar properties and modes of action, applying frameworks for grouping and read-across for nanomaterials [9].

Additionally, measuring the exposure to ENM, such as the measurement of aerosolized ENM in the breathing zone of an individual, is challenging, requiring the use of reliable and accurate portable detectors. Much effort has been made in recent years regarding the development of such instruments, the results of which are promising [10,11,12]; however, there are still certain limitations, as these instruments often report large deviations in particle concentrations, underestimate particle diameters [13,14], and generally have lower accuracy and comparability compared with their stationary counterparts [10]. Furthermore, high-hazard substances may require specific chemical particle analyses if they pose a risk at concentrations at or below the usual mass or number background concentrations.

In 1996, the American Public Health Association passed a resolution entitled, ‘The Precautionary Principle and Chemical Exposure Standards for the Workplace’, to be used when there is not enough scientific evidence to assess the safety of a material [15], as is the case for ENM. The precautionary approach is, therefore, highly recommended for risk assessments concerning ENM. 

Several risk assessment methods are available for ENM, some of which are based on control banding (see Table 1). Control banding [16] has been identified as a pragmatic strategy for grouping materials based on the similarity of properties when not enough detailed data are available for each material. Materials with similar structures and hazardous properties are grouped into bands and protective measures are defined by the risk levels of these bands. 

Some of these methods are primarily risk assessment tools that can be used to determine whether a process presents high or low risk (Table 1, entries 1–4). 

The decision-making framework for the grouping and testing of ENM (DF4nanoGrouping) was developed by the Nano Task Force to reduce the testing necessary for the hazard assessment of nanomaterials [17]. Stoffenmanager Nano evaluates health risks based on available information, for example safety data sheets. The user-friendliness of their approach has been tested and reviewed by companies in the field [18]. The Federal Office of Public Health (FOPH) and Federal Office of the Environment (FOEN) in Switzerland developed the Precautionary Matrix for the self-control of industry, commerce, and trade when dealing with synthetic NM (FOPH guidelines). The MARINA risk assessment strategy provides guidelines for assessing risks involved with ENM in a two-phase process: initial problem-framing and a subsequent risk assessment [20]. 

Four of the tools propose additional mitigation measures after the initial risk assessment and are, therefore, considered risk management tools (Table 1, entries 5–8). The control banding nanotool (CB nanotool) uses a severity–probability matrix, whereby a series of criteria are evaluated to give a score [21,22,23]. The French Agency for Food, Environmental, and Occupational Health and Safety (ANSES) developed a control banding tool for managing nanomaterial risk in individual work places [24]. The tool is made to be used by chemical safety specialists with some background knowledge of ENM and nanotoxicology. The National Research Center for the Working Environment in Denmark developed Nanosafer for risk management in specific work scenarios [25]. 

The EPFL method, presented as a product of the work group Nanosafe, is a method used for classification of nanomaterial activities based on exposure assessments [27], and particularly on hazardous materials, namely carbon nanotubes (CNT). The method, based on both the precautionary principle and control banding, was updated in 2016 with the introduction of a first hazard assessment step that was then followed by an exposure assessment based on the emission potential of the activity [26]. 

The EPFL classification method was tested in close collaboration with ENM research groups at EPFL, which demonstrated the added value of the initial hazard assessment and additionally unearthed certain gaps in the method [28]. 

The main focus of this study is related to the exposure assessment and consequent nanoclassification of the laboratories with the development and determination of threshold values. At the same time, the identified gaps in the hazard assessment were grouped into seven types of nanomaterials that are regularly used in research: Inorganic materials with organic coatings;Organic polymer materials;Biomaterials;Noble metal ENM;Metal alloys;Carbon-based materials;Nanofibers.

We performed a substantial literature review to determine the hazardous properties of these materials in order to group them into hazard bands. Herein, we present an updated hazard assessment method that now includes these new types of ENM. 

These modifications were required to better adapt the methodology to actual situations encountered in the laboratories and to simplify the classification process to make it a useful tool for researchers. For this purpose, the survey on the use of ENM at EPFL performed in 2017 has been essential for collecting ideas and information [28].

## 2. Hazard Assessment

Based on the feedback from widespread classification campaigns on the EPFL campus, three types of ENM were identified as incorrectly placed in hazard bands, i.e., biomaterials, organic polymer materials, and inorganic materials with organic coatings. These materials were all systematically placed in the H2 hazard band because none of the questions in the decision tree reflected their properties, which obliged the user to answer all the questions with “No” or “I do not know”. 

In addition to the new hazard classes, some of the previously classified material classes needed to be updated to reflect state-of-the-art studies, i.e., nanofibers, graphene-based materials, noble metal nanomaterials, and materials capable of generating reactive oxygen species (ROS). These question branches were modified based on recent studies. If the hazard cannot be estimated, a full hazard assessment must be performed to determine the hazard level of the ENM. 

The hazard assessment tree presented herein consists of 10 sections with yes/no/I do not know questions that lead the user to a hazard class for the ENM (Table 2). Within each section, the properties of the material are assessed with a branch of questions that end with the appointment of a hazard level (H1, H2, H3) for the material (Figure 1 and Figure 2).

### 2.1. Section (a): Safety Data Sheets (SDS)

The safety data sheet for a chemical provides the user with basic information on the hazards and safety measures. It is the most accessible safety information available for chemicals, and it is a legal obligation of the supplier to also provide the corresponding SDS. 

SDS are provided with ENM bought from a supplier, but they are often based on the corresponding bulk material without being modified to reflect the nano-specific properties, such as the size-dependent and physicochemical properties. Eastlake et al. developed a scoring system with four criteria for the evaluation of ENM SDS and used it to evaluate their reliability [30]. Following the labelling regulation introduced in 2012, the group showed that out of 67 SDS used for commercial nanomaterials, 36% of the SDS were unreliable in terms of ENM information and 36% were partly reliable [31].

The Swiss State Secretariat for Economic Affairs (SECO) prepared a guide to help manufacturers correctly complete the safety data sheet for nanomaterials [29]. The guide highlights particularly important information for nanomaterial SDS. We modified the question regarding existing regulations to only refer to SDS that comply with the SECO recommendations. The SDS remain some of the most important safety information documents for any commercially purchased chemical compound, and we encourage the extraction of any possible information from the SDS as a first safety measure. 

Figure 3 outlines the use of GHS pictograms and hazard statements to place materials in hazard bands.

### 2.2. Section (b): Already Classified Materials or Components

This section involves materials that contain a component that is classified as H3, such as synthetized materials that lack official hazard classification, as well as certain alloys. Based on the precautionary principle and the possible leaching of ions, atoms, or molecules from surfaces, these materials are placed in the H3 hazard group. Examples are heavy-metal-containing materials, such as lead perovskite ENM and ENM coated with toxic molecules. 

### 2.3. Section (c): Nanofibers 

The potential hazard of nanofibers has been related to the fiber paradigm, based on their structural similarities to micron-sized fibers such as asbestos. The most important factors that define the impacts of fibers present in the lungs are their length, thickness, and biopersistence [32,33]. Fibers shorter than 5 µm can be eliminated by the lung macrophage system, while longer ones remain in the lungs and lead to frustrated phagocytosis and permanent inflammation [34]. 

Studies have shown that CNT cause inflammation at relatively low aerosol concentrations [35,36]. Additionally, metal atoms on the surface of the CNT—possible residues from fiber synthesis—can further aggravate the damage via increased ROS formation and oxidative damage to cells. 

The nanofiber category contains materials that are longer than 5 µm, have a diameter of less than 3 µm, and an aspect ratio of at least 3:1. These materials are known under many names. Below is a non-exhaustive list of materials that are included in the nanofiber category:Nanofibers;Nanowires;Nanotubes;Nanorods;Nanowhiskers;Nanocylinders;Nanotubules;Nanosprings;Nanoropes;Nanofilaments;Nanofibrils.

This section starts with the determination of the biopersistency of the nanofibers. Non-persistent fibers are classified as H2, no matter their length, since fiber-related chronic effects can be excluded due to the elimination of the fibers. For biopersistent fibers, the length is the determining factor. Fibers longer than 5 µm are categorically classified in the H3 band, since they have the ability to cause considerable harm in the lungs upon inhalation. 

Because it is difficult to control the fiber length during synthesis, and in line with the precautionary principle, fibers shorter than 5 µm are only classified in the H2 band if they have been produced in a controlled Good Laboratory Practice (GLP) process that guarantees that none of the fibers are longer than 5 µm [37]. 

### 2.4. Section (d): Polymers and Biomaterials 

The biomaterial category includes ENM from biological sources [38]. Biomolecules, even if they are often in the nano size range, are parts of living systems and are dealt with in the biohazard assessment. We, therefore, start the line of questioning by removing biomolecules as non-nano-relevant. 

Biodegradable polymer ENM are often used in drug transport approaches and are designed to be relatively safe [39]. They are recognized by phagocytes and are subsequently eliminated [40]. This group of materials includes micelles, dendrimers, liposomes, and similar carrier molecules.

Most plastic polymers are biochemically inert and can, therefore, be considered fairly harmless for humans, although residual monomers from incomplete polymerization can cause health issues depending on the monomer properties [41]. Synthetic polymers that degrade under physiological conditions are, therefore, classified based on the toxicity of the monomer or sub-chains using table GHS. 

Although biological capping agents generally reduce the toxicity of ENM [42], coated polymer ENM could not be grouped into one hazard band because of the immense possible variations in composition. We, therefore, propose that a case-by-case risk assessment be performed for each coated polymer nanomaterial. 

The effects of surface charges have been studied, although the results remain inconsistent [43]. Some studies have indicated that positively charged particles behave differently in biological systems [44]. On the whole, negatively charged particles are generally more stable in serum media, whereas positively charged particles tend to aggregate due to bridging flocculation of their positively charged surface with negatively charged proteins [45]. Positively charged polyelectrolyte ENM, on the other hand, have been shown to cause red blood cell (RBC) aggregation and cytotoxicity in vitro [46]. These materials were, therefore, placed in the H2 hazard band. 

### 2.5. Section (e): Pure Carbon Materials 

The pure carbon materials class was revised after an updated literature study. Carbon black was previously placed in the H1 hazard band, although was upgraded to the H2 hazard band. IARC classifies carbon black as possibly carcinogenic to humans, with sufficient evidence in experimental animal trials (group 2B) [47], although no link has been made between occupational exposure to carbon black and cancer in humans [48]. Several studies have indicated that carbon black in nano-form is more reactive and also more toxic than the bulk form. 

The term “graphene” includes several types of graphene-based materials, some less hazardous than graphene itself. These materials were thoroughly evaluated in a safety assessment as part of the Graphene Flagship project [49]. In this project, a thorough literature review was performed to assess the human and ecological hazards of graphene-based materials. The primary exposure in occupational settings is through inhalation, so the main adverse event is pulmonary damage. The review concluded that while there is an initial immune response with an initial spark of inflammation in the first days after exposure to graphene-based materials, very few studies indicate pulmonary fibrosis or chronic toxicity [49]. It was, therefore, concluded that graphene-based materials can be classified in the H2 hazard band. 

### 2.6. Section (f): Solubility 

In this section, the solubility of ENM in near-physiological conditions is considered. When an ENM is dissolved, the toxicity depends of its chemical components and chemical toxicity, while insoluble particles can express particle toxicity or nanotoxicity. 

In the previous version of the hazard assessment tree, a solubility threshold of 0.1 g/L was adopted, as given in [50], analogous with the value for practically insoluble drugs defined in the European Pharmacopoeia [51]. An ENM with a diameter of 10 nm is considered the same as for the bulk material, while the solubility of ENM with diameters ranging between 1 and 10 nm depends more on the size; smaller particles are generally more soluble than large particles, as described in the Ostwald–Freundlich equation [52].

When an ENM is dissolved it forms dissolved ions, molecules, and atoms, whose toxicity has already been established. The H level of these materials equals their GHS classification. The non-soluble nanomaterials are further assessed based on their nanomaterial properties. 

### 2.7. Section (g): Inorganic Material with Organic Coating

Coating an ENM with an organic coating can change its properties significantly [53,54]. The effect of a coating on an inorganic nanomaterial depends on its stability and inherent functionalities. A coating can affect the size, solubility, and behavior under the physiological conditions of an ENM. If the coating is not stable under physiological conditions, it dissociates when entering the body and is replaced by a protein corona. The particle hazard in that case depends fully on the properties of the inorganic core. 

If the coating is stable on the particle, it likely has a detoxifying effect on the core, as long as the coating is not toxic in itself. The next question determines whether the inorganic core is toxic or is made up of toxic elements, in which case the ENM is placed in the H3 band. If the inorganic core is not H3 and the coating is biocompatible, the ENM is classified as H1, while for incompatible coatings or when there is not enough information to determine that the coating is biocompatible, the ENM is classified based on the hazard level of the inorganic core. 

Whether the coating material stays on the particle or not can be investigated by comparing the z-potential of the core material with the coated ENM [55]. 

### 2.8. Section (h): Pure Metals 

Gold nanoparticles (NP) are well known as highly biocompatible and are extensively used in medical applications. As such, they are placed in the H1 hazard band. In the previous version of the hazard assessment tree, gold NP with a diameter below 10 nm were classified as H3 because of their ability to enter the cell nucleus and interact with DNA [56]. 

Following this paper, many studies were performed with different gold NP. In general, gold NP are highly compatible with biological environments and are often developed for medical use. Larger gold NP with diameters above 3 nm are inert, non-toxic, and rapidly eliminated by the kidneys or the liver [57]. 

On the other hand, ultra-small gold NP (diameter <3 nm) have been shown to affect cell viability and differentiation in vitro [58], to dissolve and reassemble inside cells [59], and can distribute to and accumulate in healthy tissue [60]. 

On the other hand, May et al. [61] showed that while there is initial DNA damage when exposing cells to small gold NP (3–4 nm), these effects are transient and the DNA lesions are readily repaired. Vales et al. [62] indicated that the toxic effects depend on the charge of the coating rather than the size of the gold NP. 

While more research is necessary to conclusively determine the toxicity of ultra-small gold NP, given the more recent findings, gold NP smaller than 10 nm are placed in the H2 hazard band instead of in H3, as was previously the case. The following question places the remaining gold NP, i.e., those with a diameter larger than 10 nm, in the H1 band. 

The platinum group metals have shown great promise as nanozymes—nanomaterials with enzyme-like characteristics. Platinum, palladium, and iridium nanoparticles show great promise in medicinal applications. They are used in diagnostic imaging and radiation therapeutics because of their near-infrared (NIR) absorption properties [63,64]. In vitro and in vivo studies have indicated the low toxicity of these ENM, but questions remain regarding their accumulation over time and what concentrations can be tolerated before this would be an issue [64]. Similarly, they have been investigated for radiation applications, with initial biocompatibility testing indicating low in vitro toxicity, contrary to Ir(III) ions, which show dose-dependent effects [65]. Ruthenium and rhodium are often used in catalysis for water splitting and hydrogenation reactions because of their high surface energy and high stability. Ruthenium nanoparticles show antibacterial activity against Gram-positive and Gram-negative bacteria [66]. 

While platinum nanomaterials show high biocompatibility and low toxicity, a few studies have indicated possible genotoxicity caused by platinum NP [67]. Additionally, the extent of accumulation is not completely understood yet. This group of materials is, therefore, placed in the H2 hazard band. 

Finally, the hazard level of an alloy depends on the individual components, and it cannot be excluded that atoms or ions leach from the material. Alloys containing hazardous components (H3) were already placed in a hazard band in Section 2.2, while for the remaining ones a hazard assessment based on literature search for each classification of a metal alloy is proposed. 

The remaining metals are classified in the H3 hazard band because of the high probability of toxic ion release, increased ROS formation, and oxidative stress. Examples of ENM that are placed in this hazard band are Ag, Zn, and Cu, which all show exceptionally high degrees of cytotoxicity [68,69,70,71,72,73,74,75].

### 2.9. Section (i): Metal Oxides and Semiconductors

Metal oxides and semiconductor materials are treated in the last part of the decision tree. Amorphous materials are classified in the H1 hazard band, with the exception of silica, which is classified as H2. 

The remaining crystalline materials are evaluated based on their ability to generate ROS. ENM have been shown to increase ROS production, and as a result cause oxidative stress [76]. Metal oxides are capable of acting similarly to nanozymes, with reactivities similar to metal ions, and have been shown to display ROS-regulating activity [77,78]. Materials that are known for their ability to generate ROS based on chemical reactivity or corrodibility are singled out by the question, “Is the ENM capable of generating ROS?”. 

Of special concern are materials whose conduction band energy levels overlap with the cellular redox potential from −4.12 to −4.84 eV [79]. Two selection criteria are used to evaluate this capability, the band gap energy and the energy of the lower edge of the conduction band. For materials with a band gap energy in the energy range of visible or near-infrared light (from 3.16 eV at a wavelength of 400 nm to 1.55 eV at 800 nm), the excitation of the electrons in the valence band during handling under daylight conditions could be possible, increasing the photoreactivity of the particles. Since this is correlated with increased ROS production, these materials are placed in the H2 hazard band [26]. 

Once the materials have been assigned into hazard bands, the acceptable exposure to the used materials is evaluated based on the hazard levels. 

## 3. Exposure

A new approach to evaluate exposure and for consequent nanoclassification was developed based on the total amount of ENM used in a laboratory per day. The use of a frequency–duration matrix [26] was reconsidered due to difficulties in determining the time needed in advance for new research studies. The implementation of a standard laboratory procedure requires time and many attempts, and the classification method must be flexible and able to adapt to a degree of uncertainty. The ENM state, whether as a solid or suspension, was not taken into consideration, since a leak as a fine liquid aerosol would quickly generate a solid aerosol after solvent evaporation, making no difference between the two states in terms of exposure. Indeed, as soon as there is an open container with a dispersion of ENM in a solvent, there is the possibility of developing an aerosol from the surface [80]. As previously underlined, measuring ENM is not an easy task and most studies pertain to the release measurements more than exposure [81,82,83,84,85,86]. Although there have been effort made to establish a database of measured exposure values [87], complete data sets are still missing. These measurements confirm the release of ENM in the laboratory, although the real exposure of the worker is often unknown. Moreover, every laboratory is different and there are several parameters to consider, such as the geometry of the room and the ventilation [88,89]. To overcome this lack of data, theoretical models and simulations can be helpful [88,89]. 

### 3.1. Threshold Values

Uncertainty factors are often used to derive OEL values in the absence of complete sets of data [90]. For example, the BSI guidelines for the safe handling of ENM [91] suggests applying different factors to the bulk OEL for three categories of ENM: carcinogenic, mutagenic, asthmagenic or reproductive toxin (CMAR), insoluble, and soluble. This idea was integrated in our model. Threshold values for different categories of ENM were determined based on benchmark exposure levels suggested by the BSI and a list of recommended OEL from different institutions, as well as from literature research (see Appendix A). Moreover, a threshold for ENM was established at 1% of the bulk value for the three different hazard categories (See Table 3). 

The starting point was the OEL for inert respirable dust, which is in the milligram range (3 mg/m^3^ in Switzerland (SUVA)). ENM belonging to the H1 category are considered the least hazardous, and for the bulk (size < 10μm) a cut off value of 1 mg was considered (Table 3). A factor of 0.1 was subsequently applied in order to calculate cut off values for H2 and H3 bulk materials. The BSI guidelines suggest factors between 0.066 (insoluble ENM) to 0.5 (soluble ENM) to go from a bulk to a nano OEL. In our case to consider the limitations of the mathematical model and for the precautionary principle, a factor of 0.01 was applied. These obtained “nano values” of 10 ug/m^3^, 1ug/m^3^, and 0.1 ug/m^3^ for H1, H2, and H3 nanomaterials, respectively, were comparable with the lowest recommended values discussed in the literature (see Appendix A). For example, for fullerenes (H1 material) Aschberger et al. suggested an OEL of 7.4 µg/m^3^, while for Ag (H3 material) Stone et al. suggested 0.098 μg/m^3^.

### 3.2. Emission Simulations

Several mathematical models can be applied to estimate exposure values, and as a first approximation for our classification method, the well-mixed room model was chosen. This model lacks accuracy in the near field [92], but has been successfully used in the past to simulate an accident in a laboratory consisting of a leak of ENM, highlighting the important role of ventilation in dispersing and eliminating the ENM [85].

The following simulation parameters were considered: Laboratory volume: 75 m^3^;Air renewal: 5 h^−1^;Accepted potential leak from the considered process is 10% in a worst-case scenario;Accepted potential leak from the fume hood to the surrounding area is 1%.

The 1% value was based on the hypothesis of a standard fume hood following the Good Laboratory Practices. Several experimental studies have validated this choice. Fonseca et al. [81] performed drop test experiments simulating accidents involving much higher amounts of nanoparticles (from 5 to 125 g) than the quantities handled in an average research lab (milligrams range). They showed that in these conditions, a fume hood with an adequate sash height and face velocity prevents 98.3% (median value 99.8%) of particle release on average, with a total range of 77.8 to 99.9%. Even a local exhaust ventilation system, when properly used, was found to reduce nanoparticle exposure by 96% for a reactor used to make nanoscale-engineered metal oxides and metals [93]. The emissions from furnaces used for the production of CNT were also efficiently captured by a fume hood [83], although the release of ENM outside the fume hood was observed under specific circumstances, such as increased circulation of airflow between the researcher and the hood or inappropriate face velocity. ENM handling enclosures and biological safety cabinets also showed excellent performance in terms of reducing emissions [94]. 

### 3.3. Definition of Nano Levels

The 8 h exposure was simulated for ENM handled in a fume hood for quantities ranging from 0.3 mg to 30 g. These calculated values were compared to the threshold H values reported in Table 3. The threshold N values for nano 1, nano 2, and nano 3 labs were set at exposure values that were 10 times smaller than the ones in Table 3, in order to guarantee the safety of the user (see Table 4).

As an example, for ENM belonging to hazard level H1, a quantity of 3 g handled in a fume hood corresponds to an estimated 8 h exposure of 1 µg/m^3^. It is possible to increase this quantity up to 30 g before reaching the threshold limit of 10 µg/m^3^ in 8 h. Considering these estimated values and by applying a factor of 10 each time, it is possible to obtain the following thresholds for the different nano levels:>30 g: Nano 3, with 30 g corresponding to an exposure of 10 µg/m3 in 8 h if working under a fume hood;3 g < X ≤ 30 g: Nano 2;≤3g: Nano 1, with 3 g corresponding to an exposure of 1 µg/m3 in 8 h if working under a fume hood.

Similar considerations were made to determine the thresholds for H2 and H3 materials. Regarding fibers, an extra 10-fold factor as a safety precaution was applied, starting from the threshold for H3 materials. These values were used to build Table 5, which allows the nanoclassification of a laboratory based on the hazard level of the ENM and the total amount handled per day.

Three special cases were identified, for which the classification table does not apply. The first case is when all ENM are exclusively handled in a fully confined environment, such as a glovebox or equivalent. In this case, the laboratory is classified as nano 1, since the risk of exposure is negligible. A second case is when ENM are embedded in a solid support, meaning a solid material (crystalline or amorphous), such as a polymer, metal, or ceramic, where there is no chemical reaction between the solid support and the ENM. If there is no possibility to release powder, the laboratory is classified as nano 1, otherwise the classification table has to be applied. The third case corresponds to a process that can release more than 10% of aerosols. Examples of processes with a high potential for emissions are pulverization, spraying, sonication, sanding, and other cases producing similar amounts of emission, such as: Transfer of brittle material, powder, liquid, or gas;Preparation of dispersion;Drop casting;Deposition;Spin coating;Centrifuging;Printing;Microfluidic spray drying;Sonication (bath or tip);Lyophilization;Nebulization;Aerosolization;High energetic transfer of liquids.

Other processes that can release ENM:Weighing powder;Handling powder;Chemical vapor deposition furnaces;Machining, abrasion, or milling.

In these cases, and others producing similar emissions, a risk assessment is required. 

Concerning the use of ENM outside a fume hood in nano 1 laboratories, a risk assessment is required for H1 and H2 ENM, while these conditions are not authorized for H3 materials. Working with nanofibers in a nano 1 laboratory, even under a fume hood, is not authorized.

Many nanomaterials have similar health effects, except at different dose levels. In the absence of specific toxicokinetic interactions, the concept of dose addition should be applied [95]. If ENM belonging to different hazard levels are used in the same laboratory, the sum of the relative quantities must be less than 1 for the considered nano class.
(1)$$\frac{X_{H1}}{{\mathrm{Max}}_{H1}}+\frac{X_{H2}}{{\mathrm{Max}}_{H2}}+\frac{X_{H3}}{{\mathrm{Max}}_{H3}}\leq 1$$
X_H1,H2,H3_ = quantity of ENM belonging to the hazard level H1, H1, H3 handled in the lab

Max_H1,H2,H3_ = maximum tolerable quantity for H1,H2,H3 materials for the considered nano class (e.g., for Nano 1: Max_H1_ = 3 g, Max_H2_ = 300 mg, Max_H3_ = 30 mg)

## 4. Preventive and Protective Measures

The preventive and protective measures corresponding to the different nano laboratory classes are presented in Table 6, Table 7, Table 8 and Table 9. Safety measures are grouped as technical (Table 6), organizational (Table 7 and Table 8), and personal (Table 9). A set of specific measures is recommended for each nano level.

### 4.1. Nano 1

Nano 1 laboratories are characterized similarly to conventional chemical laboratories, with some additional specificities for handling ENM, such as an air renewal rate of 8 times per hour and a tiled or sealed floor that is easily cleaned to avoid contamination. Personal protection is the same as that of a conventional chemical laboratory (see Table 9). Gloves are selected according to their compatibility with the chemicals and solvents used in the experiments, with a few additional material recommendations specific for ENM, according to permeability studies for certain categories: nitrile for CNT; nitrile and neoprene for TiO_2_ and Pt; nitrile, neoprene, and vinyl for graphite [96,97]. 

### 4.2. Nano 2

The protective measures for nano 2 laboratories described in Table 6 state the same air exchange requirements as nano 1, but with a higher negative pressure in the nano lab (for nano labs 2 and 3, the negative pressure must be controlled). There is an obligation to work under a fume hood with an average face velocity ≥ 0.3 m/s over the number of points measured according to EN 14175-3 [98] (e.g., n = 12 for a 1200-mm-wide fume hood), with a relative standard deviation <15%. Access is restricted to trained and authorized personnel and adapted training must be provided to cleaning and maintenance personnel. Additional personal protective measures include a non-woven, disposable lab coat, which is discarded at the end of a workday after heavy use, or at least once per week, and immediately if contaminated with ENM. 

### 4.3. Nano 3

The highest safety level nano 3 contains the most important protective and preventive measures, as depicted in Table 6, Table 8 and Table 9. At this safety level, there is a strengthening of measures in all areas. 

The ventilation is increased to an hourly renewal rate of 10 times per hour and the rejected air is filtered by an H14 type filter [99]. An H14 filter has a 99.995% average efficiency for 0.3 μm particles and is used to avoid any dispersion of ENM to the environment. A negative pressure of 20–25 Pa is applied to the laboratory and an airlock with a safety shower involving recovery of drain water is mandatory inside. This allows clear segregation between corridors and nano laboratories.

Access is restricted and strictly controlled, for example with a local or remote access control. A register is kept to ensure the traceability of people who enter the laboratory. Cleaning of the room is performed by the regular laboratory staff. All regular laboratory staff, as well as people exposed to ENM, are subject to medical surveys. 

High-protection personal protective equipment is required. An integral non-woven coverall with a hood with assisted ventilation (either air supply or P3 filter (EN143) [100]) is recommended, together with two long pairs of adapted gloves. 

### 4.4. General

The organizational measures common to all nanoscale laboratories relating to training, reception and shipping, transport, and disposal, as well as the protection of pregnant women, are summarized in Table 7. The measures for transport and disposal aim to reduce the accidental dispersion of ENM into the environment, preventing further contamination. 

An additional risk assessment is proposed for pregnant women to avoid any potential complication from exposure to ENM. Work authorization is given by an occupational physician experienced in ENM risk assessment. 

Nano 1 laboratories can be cleaned by regular laboratory cleaning staff, although for nano 2 laboratories additional nanomaterial hazard training is necessary. In Table 8, the regular cleaning staff are referred to as external staff and qualified external personnel are part of the regular cleaning staff to whom specific training relating to nano 2 laboratories is given. Only wet cleaning techniques should be used in ENM laboratories. If a vacuum cleaner is used, it must be done with an asbestos type cleaner equipped with an air filter at the exhaust. 

Maintenance procedures require particular attention. It has been observed that this aspect is frequently neglected and that maintenance personnel are exposed to hazards. For example, a glove box must be decontaminated before opening it for maintenance or intervention. The contained air must be filtered through a minimum H13 filter [99]. These procedures and protocols relate to both infrastructure and research equipment. The maintenance procedures are the same for all nano levels, although for nano 2 and nano 3 maintenance, nano 3 personal protective equipment must be worn.

Table 8 outlines general recommendations for the maintenance and cleaning of nano laboratories, which do not account for the specific regulations of any institution. In addition to the recommendations given herein, each institution should develop specific procedures for maintenance, cleaning, and accident response.

## 5. Conclusions

Herein, we have presented a comprehensive and easy-to-use tool for risk assessment when working with engineered nanomaterials. The tool is destined for laboratory research and is particularly helpful when planning new research and setting up an ENM laboratory. After an initial hazard assessment of the ENM, which is performed using a decision tree with a series of yes/no/I do not know questions, the risk level of the laboratory is determined based on acceptable exposure scenarios. 

The probable exposure is estimated using a “well-mixed room” model and a maximal emission level of 10% of the used nanomaterials. The laboratories are then divided into three risk bands, herein called nano levels, with each nano level assigned a set of protective and mitigation measures. 

The comprehensive measures cover technical, organizational, and personal protective aspects based on the estimated risk. The measures are designed to protect the workers and the environment from hazardous exposure to ENM.

## Figures and Tables

**Figure 1 nanomaterials-11-02768-f001:**
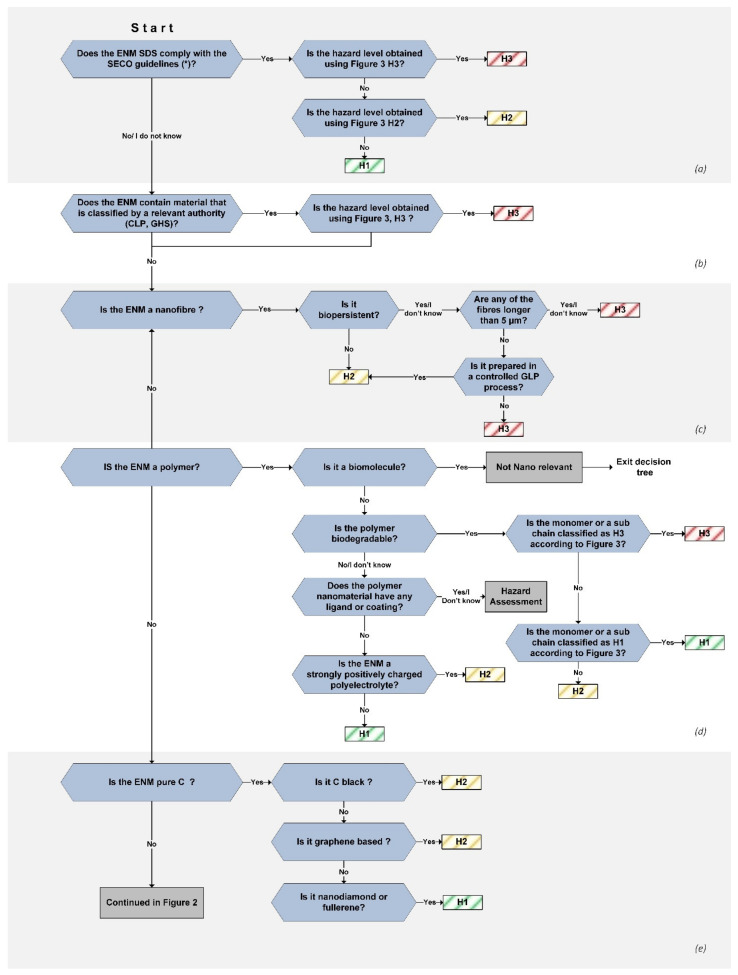
Decision tree used for the hazard assessment of nanomaterials in groups (**a**–**e**). The decision flow continues in the decision tree in Figure 2. The assessment is made by going through the yes/no/I do not know questions from the beginning until an H level has been assigned. * SDS Guidelines for synthetic nanomaterials determine nano specific information necessary for each section of the SDS [29].

**Figure 2 nanomaterials-11-02768-f002:**
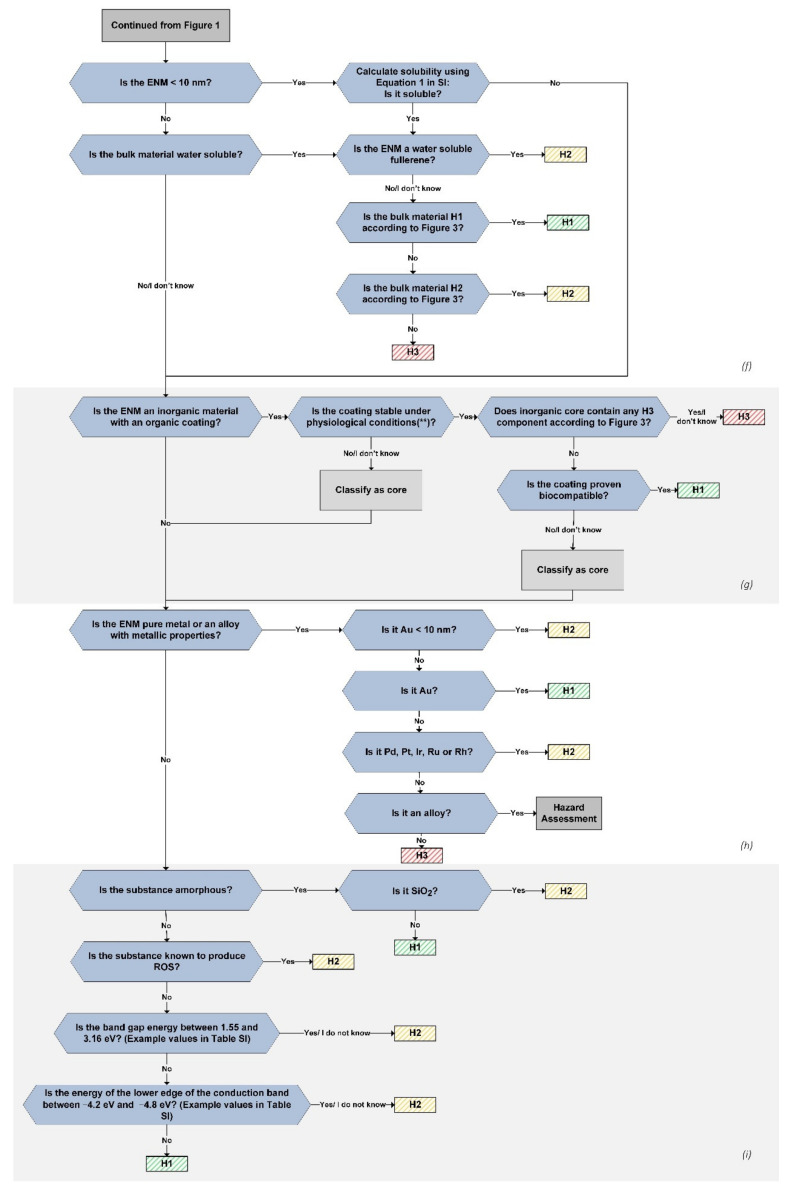
Decision tree used for the hazard assessment of nanomaterials in groups (**f**–**i**). The decision flow continues from the decision tree shown in Figure 1. The assessment is made by going through the yes/no/I do not know questions from the beginning until an H level has been assigned. ** In lungs: pH 4.5 – 5 after complete phagocytosis, pH 7.4 in extracellular fluid.

**Figure 3 nanomaterials-11-02768-f003:**
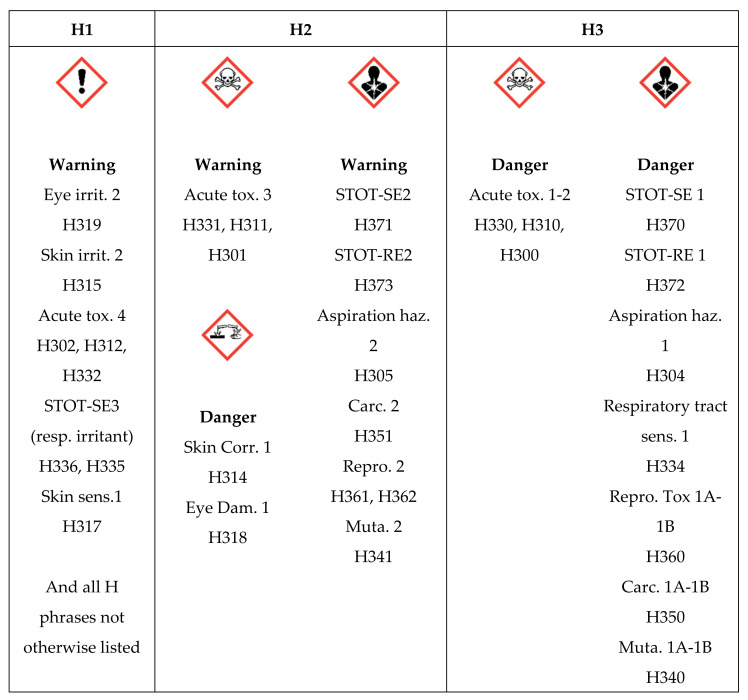
Classification of ENM into three hazard levels based on the Globally Harmonized System of Classification and Labelling of Chemicals (GHS).

**Table 1 nanomaterials-11-02768-t001:** Summary of existing risk identification (RI), risk assessment (RA), and risk management (RM) methods for the occupational use of nanomaterials.

Entry	Name	Aim	Method
1	DF4 NanoGrouping [17]	RA	Functionality-driven method that expects similar materials to behave the same. Grouping of materials is based on intrinsic material properties and system-dependent properties.
2	Stoffenmanager Nano [18]	RA	Qualitative assessment of occupational health risks from inhalation exposure to ENM. Uses product-specific information from the SDS.
3	Swiss precautionary matrix [19]	RI	Scoring system that divides materials into two risk classes. Material properties, physicochemical, and toxicological information used for scoring.
4	MARINA [20]	RA	Problem-framing with subsequent RA. Addresses four central themes for the risk assessment and management of NMs: materials, exposure, hazard, and risk.
5	CB nanotool [21,22,23]	RM	Severity probability matrix. Material properties and probability of exposure based on process.
6	ANSES CB tool [24]	RM	Control banding based on hazard level of ENM and emission potential of process.
7	Nanosafer [25]	RM	Score-based hazard and exposure assessment. Necessary information in technical and safety data sheets from supplier.
8	EPFL tool [26]	RM	Risk assessment based on material properties and emission potential of process. Mitigation measures proposed for each risk level.

**Table 2 nanomaterials-11-02768-t002:** Sections of decision tree used for hazard grouping of ENM.

Categories of ENM in Decision Tree
(a) Safety data sheet (SDS)
(b) Already classified materials or component
(c) Nanofibers
(d) Polymers and biomaterials
(e) Pure carbon materials
(f) Solubility
(g) Inorganic material with organic coating
(h) Pure metals and alloys with metallic properties
(i) Metal oxides and semiconductors

**Table 3 nanomaterials-11-02768-t003:** Threshold quantities used to determine the nano level for each hazard level.

Hazard Level	Threshold H	
	Bulk [μg/m^3^]	Nano [μg/m^3^]
H1	1000	10
H2	100	1
H3	10	0.1

**Table 4 nanomaterials-11-02768-t004:** Thresholds for nano levels.

Nano level	Threshold N
Nano 1	Simulated exposure < 1/10 threshold H ^1^
Nano 2	Simulated exposure < threshold H ^1^
Nano 3 ^2^	Simulated exposure > threshold H ^1^

^1^ Threshold H for the corresponding hazard level. ^2^ In nano 3 laboratories, respiratory protection is required.

**Table 5 nanomaterials-11-02768-t005:** Determination of nano level based on amounts handled of materials belonging to different hazard levels.

**Starting Information**				
Quantity of ENM handled Hazard level of each ENM				
**Special Cases**				
Are ENM exclusively handled in a confined environment? If yes, Nano 1. Are ENM embedded in a solid support? Nano 1 if there is no possibility of releasing powder, otherwise go to the classification table. Can the process release more than 10% of aerosols (e.g., pulverization, spraying, sonication, sanding)? Perform a risk assessment.				
**Nanoclassification**				
**Hazard Level**	**Total Daily Amount per Lab**			
	**Nano 1** **(outside Fume Hood)**	**Nano 1**	**Nano 2**	**Nano 3**
H1	Risk assessment	≤3 g	3 g < X ≤ 30 g	>30 g
H2	Risk assessment	≤300 mg	300 mg < X ≤ 3 g	>3 g
H3	Not recommended	≤30 mg	30 mg < X ≤ 300 mg	>300 mg
H3 nanofibers	Not recommended	Not recommended	≤30 mg	>30 mg

**Table 6 nanomaterials-11-02768-t006:** Technical measures for nano laboratories.

Technical Measures		Nano 1	Nano 2	Nano 3
Ventilation	Renewal rate without recycling	8 h^–1^	8 h^–1^	10 h^–1^
	Exhaust air filtration close to the source H14, standard maintenance ^1^			x
	Negative pressure between the room and the corridor ^2^	x	10–15 Pa	20–25 Pa
	Capture at source ^3^	(x)	x	x
Floor	Tiled or sealed floor	x		
	Sealed floor		x	x
Manipulation under fume hood	Recommended	x		
	Mandatory		x	x
Access restriction	Restricted (control access system)		x	x
	Automatic door closer		x	x
	Register of exposed persons + presence board			x
Airlock	Airlock ^4^			x
	Safety shower ^5^			x
Use of vacuum cleaners	Asbestos category (dust class H with asbestos specification according to EN 779)	x	x	x

^1^ Filters close to the source, separate extraction. For a nano 2 lab, the filter box is pre-installed. ^2^ For nano labs 2 and 3, the negative pressure is controlled. ^3^ Source capture for level 1 if the risk is deemed unacceptable (e.g., use of powders, aerosols, large quantities). It is the quantities handled and the nature of the products that will determine the thresholds for adding ventilation at source. ^4^ Airlock flooring must be sealed and have a minimum capacity of 200 L. An anti-panic system allowing a quick exit from the airlock must be installed. A system allowing only one airlock door to be opened at a time must be installed. ^5^ The shower is installed in the airlock without a drain.

**Table 7 nanomaterials-11-02768-t007:** Common organizational measures.

Common Organizational Measures		
Training	Laboratory training	Basic laboratory course
		Specific nano safety training
Reception and shipping	Organization	Reception point: nano lab or chemical shop
	Procedure	Reception procedure
	Storage	Ventilated cabinet or room
Transport and disposal	Conditioning of ENM contaminated materials	Toxic (trash bin for toxic)
		Double bag for toxic waste (100 microns thickness)
		Storage in a sealed container
	Disposal of ENM and products	Double packaging for liquid and solid waste
	Waste and PPE disposal	Special waste treatment channel
	Transportation of ENM	Double packaging
Pregnant women	Work authorization	Only by occupational physician ^1^

^1^ Obligatory workplace audit by the occupational physician.

**Table 8 nanomaterials-11-02768-t008:** Organizational measures.

Organizational Measures			Nano 1	Nano 2	Nano 3
General	Restricted access	Authorized persons only		x	x
		Only nano activities in the laboratory			x
	City/laboratory clothes separation				x
	Procedures	Written working procedures		x	x
Audit and follow up	Audit	Only by Occupational Safety Specialists		x	x
		COSEC	x		
	Medical survey	Not necessary	x		
		Only regular laboratory staff		x	x
		All potentially exposed persons			x
Cleaning	Who?	External staff with standard laboratory hazard training	x		
		Qualified external staff with nano laboratory hazard training ^1^		x	
		Laboratory staff only			x
	How?	Cleaning	Wet process only		
		Vacuum cleaner	Asbestos type		
	Protective equipment	Laboratory standard	x		
		Same as for laboratory staff		x	x
	Supervision	Responsible of the laboratory		x	x
		Without supervision	x		
Maintenance	With possible contact with ENM	Protection equivalent to nano 3 level		x	x
	Glove box	Possibility to put a minimum filter H13 on the ventilation of the glovebox ^2^	x	x	x
	Wastes	Identical disposal of “nanowaste”	x	x	x
	Presence of lab staff			x	x
	Contactless with nano (simple repair)	PPE protection of the corresponding lab level	x	x	x
	Maintenance procedures ^3^	Established and available procedures	x	x	x
	Maintenance protocols ^3^	Protocols established and archived	x	x	x

^1^ Qualified external personnel are part of the cleaning staff to whom specific instructions or explanations relating to the laboratories are given. ^2^ A glove box must be decontaminated before opening it for maintenance or intervention. The contained air must be filtered through a minimum H13 filter. ^3^ The procedures and protocols relate to both infrastructure and research equipment.

**Table 9 nanomaterials-11-02768-t009:** Personal measures.

Organizational Measures		Nano 1	Nano 2	Nano 3
Eye protection	Safety glasses	x	x	
Respiratory protection	Mask with assisted ventilation (hood) ^1^			x
Body protection	Coverall with hood—Tyvek^®^ style			x
	Non-woven laboratory coat (Tyvek^®^)		x	
	Cotton lab coat	x		
	Overshoes		x	x
	Sticky mat (at the exit)			x
Hand protection	1 pair of adapted gloves	x	x	
	2 pairs of adapted gloves ^2^			x

^1^ If using a cartridge filter system, filter P3. ^2^ Select gloves according to their compatibility with the materials and solvents to be used. In addition, according to permeability studies for certain categories of ENMs, the recommendations are as follows: nitrile for CNT; nitrile and neoprene for TiO2 and Pt; nitrile, neoprene, and vinyl for graphite [96,97].

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
