# Peer review of "NanoSafe III: A User Friendly Safety Management System for Nanomaterials in Laboratories and Small Facilities"

_nanomaterials, 2021, doi:10.3390/nano11102768_

Round 1

Reviewer 1 Report

The paper reports on the risk assessment procedure for nanomaterials handling on a lab scale. The subject of the paper is within the scope of Nanomaterials journal. The results presented are of special importance for the authorities and scientists setting up new laboratories focused on the study of advanced engineered nanomaterials.

I have the following comments:

1) Please check the references to the tables etc. (see lines 102, 115, 175, 196, 421, 425, 467, 486).

2) It is unclear why biological molecules are attributed to nanomaterials (see line 133). Please explain.

3) Figure 2 begins with the item entitled "Continued from Figure 2". Actually, Figs. 2 and 3 are exactly the same.

4) What are the reasons for choosing the solubility threshold of 0.1 g/L? Please explain.

5) In Section 2.9, the ability of metal oxides to act as nanozymes should be at least mentioned (e.g., enormous enzyme-like activity of nano-ceria, see doi 10.1016/B978-0-12-815661-2.00008-6).

6) Some English polishing is required (see lines 219, 541 etc.)

Reviewer 2 Report

The review focused on the risk assessment of nanomaterial processes and presented an updated version of risk assessment tool for work with nanomaterials based on a three-step control banding approach and the precautionary principle. The content is very rich and it is meaningful for prevention of occupational hazards of nanomaterials. Some suggestions are given here.

  1. The introduction part is very scattered, and attention should be paid to focusing on the main background knowledge and significance.

  1. The classification or grouping of nanomaterials is very important for risk assessment. It is recommended to refer to authoritative literature.

  1. Although I understand that the risk assessment of nanomaterials is very complex, there are many things to consider. However, the flow charts in Figure 2 and Figure 3 are still complex, and it is recommended to simplify them appropriately. Can you show the main structure first and put the details in the supplementary materials?

.

  1. There are two tables 1, which should be distinguished.

  1. This manuscript provided a great deal of information. The authors should find ways to integrate these contents.
